# A Novel BCC-Structure Zr-Nb-Ti Medium-Entropy Alloys (MEAs) with Excellent Structure and Irradiation Resistance

**DOI:** 10.3390/ma15196565

**Published:** 2022-09-22

**Authors:** Zhenqian Su, Zhaodong Quan, Tielong Shen, Peng Jin, Jing Li, Shiwen Hu, Dexue Liu

**Affiliations:** 1School of Materials Science and Engineering, Lanzhou University of Technology, Lanzhou 730050, China; 2State Key Laboratory of Advanced Processing and Recycling of Non-Ferrous Metals, Lanzhou University of Technology, Lanzhou 730050, China; 3Institute of Modern Physics, Chinese Academy of Sciences, Lanzhou 730050, China; 4University of Chinese Academy of Sciences, Beijing 100049, China

**Keywords:** Zr-Nb-Ti MEAs, microstructural characterization, irradiation resistance, defects evolution

## Abstract

Medium-entropy alloys (MEAs) are prospective structural materials for emerging advanced nuclear systems because of their outstanding mechanical properties and irradiation resistance. In this study, the microstructure and mechanical properties of three new single-phase body-centered cubic (BCC) structured MEAs (Zr_40_Nb_35_Ti_25_, Zr_50_Nb_35_Ti_15_, and Zr_60_Nb_35_Ti_5_) before and after irradiation were investigated. It is shown that the yield strength and elongation after fracture at room temperature are greater than 900 MPa and 10%, respectively. Three MEAs were irradiated with 3 MeV Fe^11+^ ions to 8 × 10^15^ and 2.5 × 10^16^ ions/cm^2^ at temperatures of 300 and 500 °C, to investigate the irradiation-induced hardening and microstructure changes. Compared with most conventional alloys, the three MEAs showed only negligible irradiation hardening and even softening in some cases. After irradiation, they exhibit somewhat surprising lattice constant reduction, and the microstructure contains small dislocation loops. Neither cavities nor precipitates were observed. This indicates that the MEAs have better irradiation resistance than traditional alloys, which can be attributed to the high-entropy and lattice distortion effect of MEAs.

## 1. Introduction

Rapidly increasing energy demands and climate change concerns inevitably emphasize the role of clean energy in the future while making fossil fuel infrastructure obsolete. After thermal power and hydropower, nuclear energy has grown to become the world’s third-largest energy source [1]. Nowadays, the GIF (Generation Nuclear Energy International Forum) has already proposed the Generation IV Nuclear System, promising competitiveness and advancement in safety, economy, sustainable development, non-proliferation, etc. However, the structure materials for the reactor will withstand very harsh environments in terms of high-temperature, high-dose irradiation, and corrosion [2], posing a significant challenge to its mechanical properties and dimensional changes attribute to void swelling or creep [1]. Hence, in order to develop advanced nuclear reactor systems, research on high-performance materials with outstanding irradiation resistance and corrosion resistance is critical. Since the 1950s, the light water reactor (LWR) has been the leading type of reactor for electricity production. Because Zr-based alloys have a low thermal neutron absorption cross-section, outstanding thermal conductivity and superior corrosion resistance, they are used as major materials for fuel cladding materials in this reactor [3,4]. At temperatures exceeding 1200 °C, however, Zr-based alloys accelerate oxidation and hydrogen pick-up, causing significant embrittlement [5,6,7], such as in the Fukushima Daiichi accident in 2011. As a result, advanced reactor systems require an upgraded Zr-based alloy or alternative materials with improved accident tolerance [8].

Recently, HEAs have attracted great attention for their excellent mechanical properties [9,10], corrosion resistance [11], and radiation resistance [12,13,14,15]. Typically, HEAs are composed of five or more elements in equal or near-equal atomic ratios [16,17], such as different variants of FeCoNiCrMn alloy, also known as Cantor alloy. The outstanding performance of HEAs is attributed to some typical characteristics, including the high-entropy effect, the severe-lattice-distortion effect, the sluggish diffusion effect, and the cocktail effect. High-stability HEAs thus may be promising candidate materials for advance nuclear systems.

The effect of the atomic number, type, and grouping of alloying components on irradiation (by ions, electrons, and neutrons)-prompted microstructural evolution and property deterioration has received much interest recently. The defect development process in alloys with high chemical complexity is slowed down by improving energy dissipation as well as defect formation and migration energy, according to recent findings [18]. Although HEA requires five or more elements, it is gradually being demonstrated that high entropy effects do not necessarily dominate the alloy structure and that the number of elements does not determine performance. HEAs have fewer densities of vacancy and interstitial defect clusters owing to their enhanced vacancy-interstitial recombination in the cascade compared to traditional alloys [19,20,21]. However, the accumulation of damage in several HEAs with varying chemical complexity has been studied experimentally. For instance, Granberg et al. likewise found a significant decrease in damage gathering under delayed irradiation in NiFe and NiCoCr compared to elemental Ni, which has been demonstrated by TEM images and MD simulations [22]. Olsen et al. additionally found defect cluster sizes yet with higher densities in NiCo contrasted to Ni [23]. Yang et al. investigated the FCC CrMnFeCoNi HEA demonstrating a very stable structure when compared to 304ss and pure Ni, both have a remarkable helium-ion irradiation resilience at room temperature and 450 °C [24]. Zhang et al. firstly discovered the dislocation loops in BCC HEAs after helium-ion irradiation at 350 °C [25].

When designing HEAs to meet the requirements of the nuclear system and the formation of the BCC structure, elements with a low thermal neutron absorption cross-section and excellent mechanical properties are considered. In addition to the Zr, there is a small amount of Nb in Zr_4_ alloy which has been used as the cladding material in pressurized water reactors. Further, the addition of Ti can enhance the corrosion performance of the alloy. In view of this, novel Zr-Nb-Ti MEAs with BCC structure was designed. To replicate neutron irradiation, the MEAs were subjected to heavy-ion irradiation. The effects of irradiation on crystal structure, hardness, and microstructure evolution were studied.

## 2. Experimental Procedures

### 2.1. Samples Preparation

The three alloy ingots of Zr_40_Nb_35_Ti_25_, Zr_50_Nb_35_Ti_15_, and Zr_60_Nb_35_Ti_5_, nominal compositions of Zr, Nb, and Ti with a purity of >99.95 wt.%, were carefully weighted, followed by melting using a cold crucible levitation technique under argon atmosphere. Additionally, they were flipped and remelted 4–5 times to ensure their chemical homogeneity, which was checked by an FEI Quanta 450 SEM (Scanning Electron Microscope) equipped with EDX (energy dispersive spectrometer). Finally, it was processed into the diverse size and shape for subsequent testing using a cutting machine. After being processed, the cross-sections of the samples were mounted in resin for the surface treatment. The grinding of the surface with sizes of 120# to 2000# and polishing to the mirror-finish surface was achieved using Tripoli, intermediates, and a finishing rough. The nominal compositions of the three HEAs are listed in Table 1. For convenience, the three HEAs were designated as 1#, 2#, and 3#, respectively.

### 2.2. Heavy Ion Irradiation Procedures

The center parts of the as-cast ingots were chopped into specimens with the dimension of 10 × 10 × 1 mm^3^. They were mechanically polished using 1.5 and 0.5 μm diamond pastes after being ground with SiC abrasive paper from 200 to 3000 grit. Finally, specimens were polished for over 8 h with 40 nm colloidal silica slurry to remove the residual strain caused by grinding and mechanical polishing.

The irradiation experiments were carried out at the Chinese Academy of Science’s Institute of Modern Physics 320 kV platform for multi-discipline research with highly charged ions. The three alloys were irradiated with 3 MeV Fe^11+^ (defocused beam) at 300 and 500 °C to ion fluences of 8 × 10^15^ ions/cm^2^ and 2.5 × 10^16^ ions/cm^2^. The effect of irradiation condition on the microstructure of 1#, 2#, and 3# specimens can be investigated by comparison with each other with low and high fluences and low temperature and high temperature. To make ion irradiation doses consistent with hypothetical neutron irradiations, using the SRIM simulation with lattice and surface binding energies of the Zr, Nb, and Ti atoms set to zero, and we used the Kinchin–Pease calculation mode based on findings from previous studies [26]. Figure 1 shows the depth profile of damage and Fe ion concentration calculated by the SRIM 2008. For the displacement energies of the lattice atoms Zr, Nb, and Ti, the value is 40, 30, and 20.8 eV, respectively. Table 2 shows the results for the major factors, including fluence, temperature, peak damage, and peak dose rate.

### 2.3. Depth-Sensing Nanoindentation Methods

Hardness measurements on samples were performed at room temperature utilizing an Anton Paar nanoindenter with Berkovich diamond indenter (3-sided pyramidal tip) for test. All the tests were performed in the continuous stiffness measurement mode (CSM) [27], with a consistent stacking rate set to 0.15 nm/s and a surface methodology speed for the tip of 10 nm/s. The entrance profundity analyzed in the examples went from 0 to 1500 nm, with a greatest heap of 200 mN, and the information inside 200 nm from the example surface was disposed of because of enormous information disperses related to surface unpleasantness. For good measurable examination, each example was indented with approximately 6 to 8 indents of each example, and the normal of the outcomes was utilized in the examination. Since the actual measured nanohardness value is easily influenced by the depth of the indenter diffusion, often the actual measured depth is 4–5-fold deeper than the depth at that point, we selected the measured value at a depth of 400 nm as the actual nanohardness value based on our experience and to fit the application of the Nix–Gao model [28]. Thus, the hardness at an indenter profundity of 400 nm was picked as the ideal condition for estimating just the 3 MeV Fe^11+^ irradiation zone without the significant information disperse seen at shallower profundities.

### 2.4. Characterization and Properties Procedure

The phase structures of original alloys were detected by x-ray diffraction (XRD) using a D/max-2400 diffractometer with Cu Ka1 radiation. The XRD parameters are voltage 45 kV, current 50 mA, Cu Kα target, X-ray wavelength 0.15406 nm, scanning range 20~100°, and scanning step 5 °/min at room temperature. However, the damage layer of irradiated samples is shallow, and the conventional X-rays penetrate more deeply than the region. This leads to the obtained information coming from both the damaged and undamaged regions. Therefore, the changes in the composition and crystalline phase were investigated by using grazing incidence X-ray diffraction spectroscopy (GIXRD). Usually, the penetration depth can be calculated using the total reflection critical Angle model [29]. The microstructures in HEAs were analyzed using transmission electron microscopy (TEM). The TEM lamellae were prepared by dual-beam focused-ion beam (FIB) lift-out method, and the microstructure was observed under the bright-field image model by FEI Tecnai F20 TEM operated at 200 kV. The room temperature mechanical properties and high-temperature mechanical properties were evaluated using Instron 5565 universal testing machine (INSTRON, Norwood, MA, USA) at the strain rate of 0.01 s^−1^. The samples utilized in the tensile test were cut into canine bone shape by wire, the gauge length and cross-sectional area of the specimens were 8 mm and (1 × 2) mm^2^, separately, and the tensile fracture surfaces were observed by SEM. 

## 3. Results and Discussion

### 3.1. Microstructure and Mechanical Behavior of the Unirradiated Zr-Nb-Ti MEAs

Figure 2 shows the XRD patterns and optical images of these three MEAs. It can be seen from Figure 2a that Zr-Nb-Tix MEAs (Zr_40_Nb_35_Ti_25_, Zr_50_Nb_35_Ti_15,_ and Zr_60_Nb_35_Ti_5_) have five diffraction peaks of BCC structure, including (110), (200), (211), (220) and (310). The strongest peaks are located on the (110) crystal plane and no other diffraction peaks appear, which indicates that the three alloys had a single BCC structure with a lattice constant of 0.3351, 0.3395, and 0.3422 nm, respectively. The diffraction peaks shift gradually towards the smaller 2θ values, which indicates that the lattice constant increases with Zr content due to the atomic radius of Zr being larger than that of Nb and Ti, resulting in lattice expansion. Using empirical formula to calculate the lattice constant of these three alloys is 0.3410, 0.3430, and 0.3448 nm, respectively, which is consistent with the values of the experimental. The SEM images of the as-cast samples reveal a homogeneous dendrites microstructure.

Several metrics for predicting the structure stability and phase formation of HEAs have been developed thus far [30]. Zhang et al. suggested a criterion for the formation of solid-solution phases in HEAs based on the enthalpy of mixing (ΔH_mix_) and atomic radius difference (δ) [31], as well as an extra parameter was also proposed as Ω = T_m_ΔS_mix_/∣ΔH_mix_∣, where T_m_ is the average melting temperature and ΔS_mix_ the entropy of mixing of an alloy. From available data, the conditions for the creation of HEAs with a single solid-solution phase are Ω ≥ 1.1 and δ ≦ 6.6%, respectively. According to the electronic structure theory, Guo et al. [32] introduced another measure named valence electron concentration (VEC) to predict the phase stability of HEAs. According to Guo’s statistics, BCC solid-solution phases are stable since the VEC value is less than 6.87. Table 3 shows the specifications of these MEAs, which are perfectly in accordance with the aforementioned criteria.

Figure 3 shows the tensile true stress–strain curve of three HEAs at room temperature and 400 °C. Table 4 summarizes the three HEAs’ yield strength (σ_y_), ultimate strength (δ_u_), and elongation after fracture (ε_ef_, _%_). The yield strength of these three MEAs at room temperature is 945 MPa, 903 MPa, and 1028 MPa, and the strain is 17.5%, 14%, and 11%. The yield strength of these three MEAs at 400 °C is 380–400 MPa at 400 °C, the tensile strength is 390–430 MPa, and the elongation rate exceeds 20%. Notably, these three MEAs also have shown excellent mechanical properties at 400 °C, even the fracture strength was reduced by about 40%, and the elongation increased by about 1 time to 28%. The fracture strength of these three MEAs is considerably higher than that of most other FCC structures HEAs, which may indicate a single BCC phase with strong bonding inherited from the refractory elements and the high solution hardening effect of the whole-solute matrix. Meanwhile, these MEAs exhibit significant plastic strain, which is uncommon for BCC HEAs.

Figure 4 shows the SEM images of the fracture surface for these three HEAs. Obvious necking near the fracture surface can be found in Figure 4a, which demonstrates that typical plastic deformation took place before fracture. As can be observed in the SEM picture, the grain boundaries are severely deformed and reveal clear vein patterns inside the crystal grains, which are distributed as densely spaced dimples on the fracture with diameters between 50 and 100 μm. Some tearing was seen beyond grain boundaries, and the tearing ridges were heavily surrounded by small dimples less than 5 μm in size. This indicates that the alloys are all more resistant to fracture and shear deformation. All of these properties, as shown in Figure 4, indicate that the fracture behavior is that of a ductile fracture, which is consistent with the significant plastic strain.

### 3.2. XRD Analysis of the Irradiated MEAs

Grazing incidence X-ray diffraction spectroscopy (GIXRD) tests were performed to investigate the irradiation effect on the crystal structure of these three MEAs, which are shown in Figure 5. Figure 5a shows Zr_40_Nb_35_Ti_25_ MEAs exhibit BCC structure with (110) and (200) planes preferred orientation. After irradiation to a fluence of 8 × 10^15^ and 2.5 × 10^16^ ions/cm^2^ and a temperature of 300 and 500 °C, the Zr_40_Nb_35_Ti_25_ MEA remains stable BCC structure without a secondary phase. However, the intensity of BCC diffraction peaks with (200) and (211) planes decreases drastically and cannot be simply connected to the 8 × 10^15^ ions/cm^2^ irradiation at 500 °C. Figure 5b shows Zr_50_Nb_35_Ti_15_ MEAs exhibit BCC structure with (110) and (200) planes preferred orientation. After irradiation to a fluence of 8 × 10^15^ and 2.5 × 10^16^ ions/cm^2^ and a temperature of 300 and 500 °C, the Zr_50_Nb_35_Ti_15_ MEA still remains as a stable BCC structure without a secondary phase. Figure 5c shows Zr_60_Nb_35_Ti_5_ MEAs exhibit BCC structure with (110) and (211) planes preferred orientation. After irradiation to a fluence of 8 × 10^15^ and 2.5 × 10^16^ ions/cm^2^ and a temperature of 300 and 500 °C, the Zr_40_Nb_35_Ti_25_ MEA also remains stable BCC structure without a secondary phase. However, the intensity of BCC diffraction peaks with the (200) plane decrease drastically and even disappears, and peaks with the (211) plane increase seriously after the 8 × 10^15^ ions/cm^2^ irradiation at 500 °C. In terms of diffraction peaks decreasing dramatically or even disappearing, Sun et al. [33] observed that the intensity of bcc diffraction peaks decreased dramatically, or even can hardly be distinguished after helium-ion irradiation for the Al_1.5_CrFeNi HEA film. Moreover, Zhang et al. [25] studied the influence of ion irradiation on the two BCC MoNbCrVTi and MoNbCrZrTi HEAs, observing a dramatic decrease in the intensity of XRD peaks. This was attributed to the amorphous structure forming as a result of high-level stress. The decreases in diffraction peak seen in this study can be attributed to local melting and recrystallization of amorphous materials caused by thermal spikes created by irradiation [20].

The diffraction peaks of these three MEAs shift gradually towards the bigger 2θ values after irradiation, implying a decrease in lattice constant. As shown in Figure 5, the lattice constant of these three MEAs has varying degrees of reduction compared to the original material within the different irradiated conditions. The observed irradiation-caused shrink of lattice constant in our MEAs is different from the conventional alloys, such as 316 L, 304 H, and Zr-Nb alloys, of which the lattice constant expands after irradiation [34,35]. Lu et al. [36] detected a drop in the Ti_2_ZrHfV_0.5_Mo_0.2_ HEA’s lattice constant with an irradiation dose of 3 × 10^16^ ions/cm^2^. On the other hand, Zhang et al. [14] discovered the irradiation response of the CoCrCuFeNi HEA, which had a rising lattice constant with an average dose of 0.73–368.5 dpa. In this work, the decrease in the lattice constant may be ascribed to the irradiation relaxes the extreme lattice distortion that exists in the HEAs due to different atomic sizes of the solutes, eventually resulting in the lattice shrink.

### 3.3. The Nanoindentation Result of the Irradiated MEAs

To further study the performance variation of these three MEAs after irradiation, the nanoindentation hardness of unirradiated and irradiated MEAs samples was measured. Nanoindentation is a valuable technology for monitoring mechanical property changes in the small affected regions. To confirm the correctness of the experimental results, six single indents were created for each sample in this investigation. As shown in Figure 6, the average nanoindentation hardness of all the studied samples was plotted as a function of the indentation depth.

The hardness decreases along with the depth in both irradiated and un-irradiated samples due to the indentation-size effect, that can also lead to the hardness value increasing with decreasing indentation size. In our work, the specimens were prepared with vibration polishing, which produced a smooth and deformation-free surface with minimal surface irregularity. Hence, the data within 100 nm are not shown since the near-surface hardness data exhibit a large scatter due to the surface irregularities [37], and indentations depth was controlled to the same depth at 1900 nm for all materials to remove the size effect [38]. It was demonstrated that the largest region of the near-surface ion irradiation region, which is independent of the underlying unirradiated substrate, corresponds to an indenter depth of 350 nm [28]. This is due to the fact that the elastic stress fields beneath the indenter are sensitive to microstructure features that are up to 10-fold the indenter depth. In order to conduct a quantitative analysis of the nanoindentation hardness of the ion irradiation specimens, indenter depths ranging from 200 to 400 nm were utilized. Figure 6a shows that the irradiated Zr_40_Nb_35_Ti_25_ MEA sample has a larger hardness than the unirradiated sample with the increase the irradiation fluence and temperature. Figure 6b shows that the irradiated Zr_50_Nb_35_Ti_15_ MEA sample’s hardness also increases, except for the sample irradiated with 8 × 10^15^ ions/cm^2^ at 300 °C. Instead, Figure 6c shows that the Zr_60_Nb_35_Ti_5_ MEA has a smaller hardness than the unirradiated, and the irradiated for 8 × 10^15^ ions/cm^2^ at 500 °C. has no obvious variation. The magnitude error bar is attributed to scattering and difficulty in fitting the data shown in Figure 6. Figure 7 illustrates the Nix–Gao model for three MEAs. The result is the same as its indentation depth profiles.

Table 5 illustrates the results of irradiation hardening and the hardening rate on the alloy. The results show the hardness value after irradiation with a high dose and high temperature are to varying changes, some materials increased, and others decreased after irradiation, but the overall pattern is still hardening. Several factors could cause irradiation-hardening, including the radiation-induced precipitation (RIP) and the formation of dislocation loops [39]. Additionally, it has been reported that the visible dislocation loops are the main contributor to the irradiation hardening of HEAs at room temperature [40,41]. As far as we know, this is firstly showing a softening of the BCC single-solution MEAs after irradiation.

Irradiation would cause hardening, embrittlement, and even softening of structure materials, which is among the main failure modes of structural materials in nuclear power plants. The three MEAs have better hardness stability than most typical alloys, which could be due to fewer flaws caused by irradiation in MEAs.

### 3.4. Irradiation Defects of the MEAs

TEM examinations of the samples were carried out to further confirm the structural development of the samples brought on by the irradiation. The initial Zr-Nb-Ti MEAs’ TEM pictures revealed a very dense microstructure. Only a tiny number of imperfections linked to the ion milling process were found. After exposure to radiation, the cross-sectional microstructures of Zr-Nb-Ti MEAs samples are shown in bright-field (BF) TEM images in Figure 8. Figure 8a shown the sample was irradiated at 500 °C to a fluence of 8 × 10^15^ ions/cm^2^, which equates to a peak damage dose of 15 dpa, as shown in the TEM image Figure 8a. The photos clearly show the differences between the irradiation-induced faults and the reference (unirradiated) samples. In contrast to the SRIM predictions, there are more black dots in the peak damage region and the defect bands of the Zr_40_Nb_35_Ti_25_ samples are concentrated in the depth range between 450 and 700 nm. In contrast to Zr_40_Nb_35_Ti_25_ samples, which are mostly scattered at a depth of 500–800 nm, the irradiation-induced defects of Zr_50_Nb_35_Ti_15_ and Zr_60_Nb_35_Ti_5_ samples demonstrate higher-density dispersion. Materials with flaws brought on by irradiation typically have voids, dislocation loops, and chemical segregation. The dislocation loop is often the most frequent defect in materials that have been exposed to radiation, and we were unable to detect any voids or precipitates in the TEM pictures. Additionally, samples with various Zr concentrations, as well as those exposed to various irradiation effects and temperatures, exhibit the irradiation-induced defects in a variety of microstructural ways. Therefore, in the materials with various Zr concentrations, we described and examined the irradiation-induced dislocation loop. It is believed that the increase in Zr content increases the high entropy effect and lattice distortion effect of the three MEAs, resulting in a larger lattice distortion degree of the alloy and some point defects annihilation, reducing the damage degree of the alloy. The outcomes of the MEAs with various Zr contents that were exposed to various radiation circumstances may be explained by looking at the nucleation and development stages of dislocation loops. More Zr contents increased the lattice distortion effect, which is used to pin irradiation-induced the defects produced during irradiation and inhibit their migration or accumulation, according to the perspective of the defect distribution. Because of this, the doped samples had a greater defect density and smaller defect size than the MEAs samples with less Zr content. 

Figure 9 shows the morphology and size of the defect of Zr_40_Nb_35_Ti_25_ samples with different irradiated conditions. Obviously, the defects in Zr_40_Nb_35_Ti_25_ samples at 300 °C to a fluence of 8 × 10^15^ ions/cm^2^ are mainly black dots that are consist of the small dislocation loops and defect clusters. In the samples irradiated at 300 °C to a fluence of 2.5 × 10^16^ ions/cm^2^ that can see more black dots with an increment of the fluence. The microstructure of another sample irradiated at 500 °C to a fluence of 2.5 × 10^16^ ions/cm^2^ contains mainly large-sized dislocation loops and dislocation lines. The aggregation of self-interstitial atoms or clusters causes these dislocation loops, which are mostly interstitial in nature. Irradiation with 3 MeV Fe^11+^ ions triggered collision cascades and the formation of interstitial atoms. These interstitial atoms clump together to create defect clusters, which absorb more interstitials or other point-defect clusters, eventually producing dislocation loops and lines. With increasing irradiation impact and temperature, there is a modest rise in loop size and a drop in loop density.

The irradiated samples did not include any holes or radiation-induced segregation (RIS). Materials exposed to radiation at doses greater than 1 dpa frequently include voids because irradiation-induced vacancies consolidate into stable cavities. Dimensional expansion and a reduction in fracture toughness are two effects of voids. It is believed that radiation-induced segregation, which occurs when alloying elements segregate out of equilibrium near sinks, is a key cause of irradiation-assisted stress corrosion cracking. As a result of the severe lattice distortion in the Three MEAs, irradiation-induced defects may diffuse slowly, inhibiting the growth of defect clusters, and defect annihilation may take place during the transition from amorphization to crystallization under conditions of high atomic-level stress.

## 4. Conclusions

In this work, we designed and prepared single BCC Zr-Nb-Ti MEAs, and then performed the Fe-ion irradiation experiments to examine their irradiation resistance. The nanoindentation experiment and microstructural characterization indicated excellent properties of the fabricated materials in terms of irradiation resistance compared with traditional alloys. The main conclusions are summarized as follows:(1)The tensile tests showed that the MEAs have a good combination of strength and toughness. At room temperature, the yield strength of these three alloys, Zr_40_Nb_35_Ti_25_, Zr_50_Nb_35_Ti_15_, and Zr_60_Nb_35_Ti_5_, is 943, 903, and 1285 MPa, and the fracture strain is 17.5%, 14%, and 11%, respectively. At 400 °C, the fracture strain increases to 28%, 25%, and 20.5%, respectively. The fracture morphology shows that the fracture mode is the ductile fracture.(2)The nanoindentation test showed the Zr-Nb-Ti MEAs have a little irradiation hardening that increased with the irradiation fluence.(3)After irradiation, contrary to traditional alloys, the XRD diffraction peaks of Zr-Nb-Ti MEAs were shifted to the right, indicating a decrease in the lattice constant. No visible phase transformation or decomposition of Zr-Nb-Ti MEAs was observed.(4)Only dislocation loops and dislocation lines were observed in the peak damage region. This suggests the irradiation resistance of the MEAs is better than that of traditional alloys.

## Figures and Tables

**Figure 1 materials-15-06565-f001:**
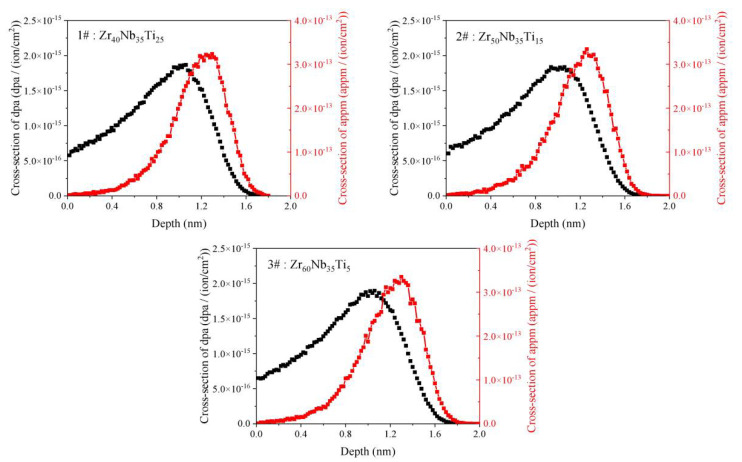
The depth profiles of damage (black line) and Fe ion concentration (red line) for the samples **1#**, **2#**, and **3#** calculated by SRIM 2008.

**Figure 2 materials-15-06565-f002:**
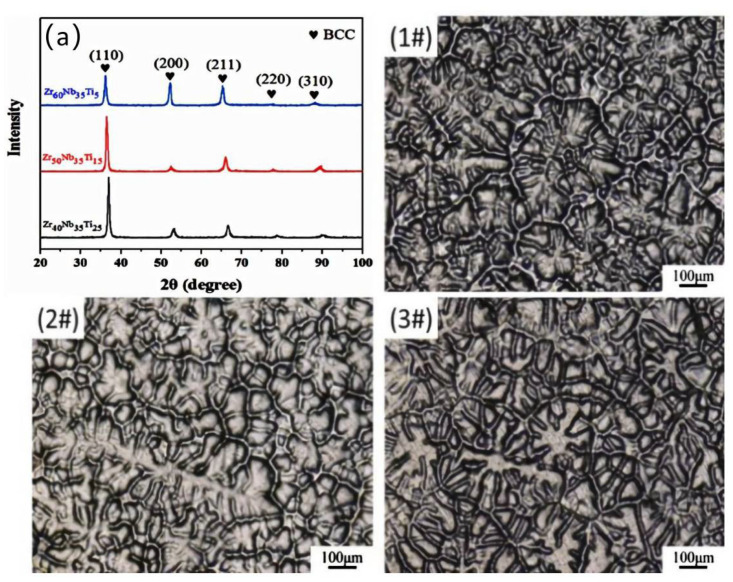
(**a**) XRD patterns and optical images of Zr_40_Nb_35_Ti_25_ (**1#**), Zr_50_Nb_35_Ti_15_ (**2#**), and Zr_60_Nb_35_Ti_5_ (**3#**) origin alloys.

**Figure 3 materials-15-06565-f003:**
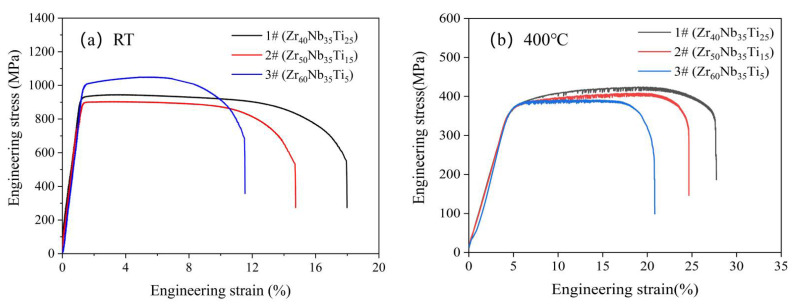
Engineering stress–strain curves of the alloys under tensile at (**a**) room temperature and (**b**) 400 °C.

**Figure 4 materials-15-06565-f004:**
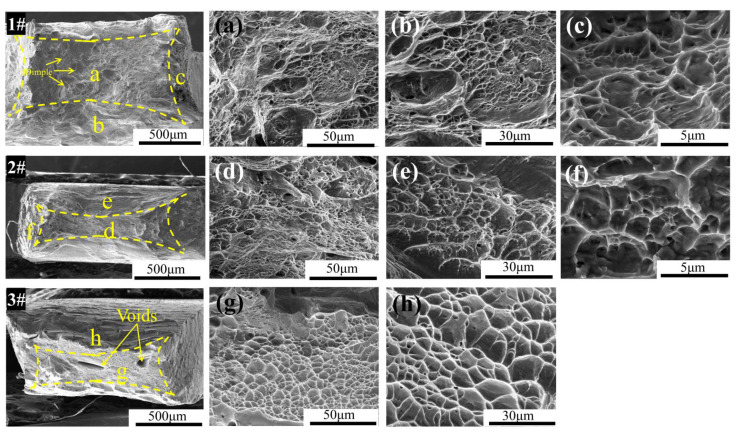
SEM images of fracture morphology for samples Zr_40_Nb_35_Ti_25_ (**1#**), Zr_50_Nb_35_Ti_15_ (**2#**), and Zr_60_Nb_35_Ti_5_ (**3#**) after tensile tests. (**a**–**c**) are enlargements of the zones labeled “a”, “b“, and “c” in the sample 1#, respectively. Correspondingly, (**d**–**f**) are the local enlargements of the sample 2#, and (**g**,**h**) is the local enlargements of the sample 3#.

**Figure 5 materials-15-06565-f005:**
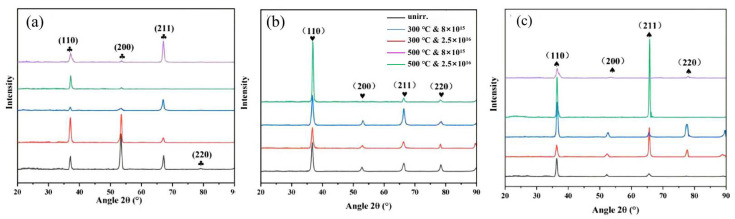
XRD patterns of the unirradiated and irradiated MEAs at different ion influences and temperatures. (**a**) Zr_40_Nb_35_Ti_25_ (**1#**), (**b**) Zr_50_Nb_35_Ti_15_ (**2#**), and (**c**) Zr_60_Nb_35_Ti_5_ (**3#**).

**Figure 6 materials-15-06565-f006:**
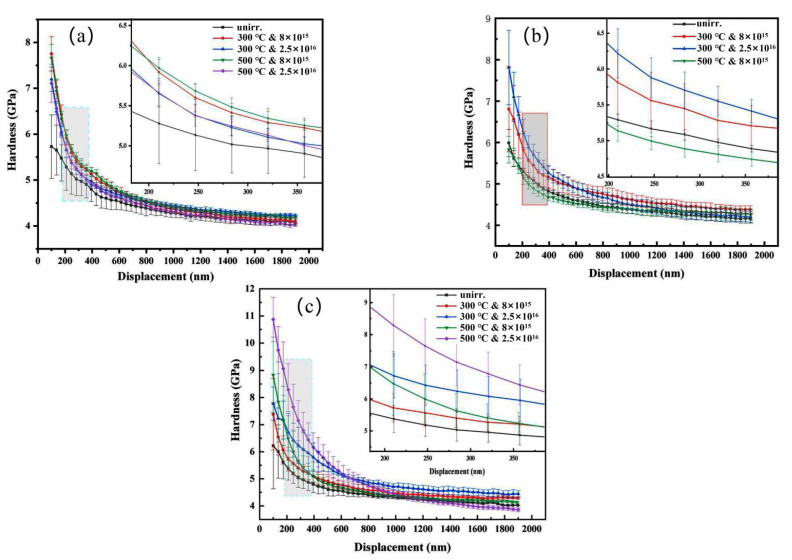
Variation of hardness with indentation depth before and after Fe^11+^ irradiation at different ion influences and temperatures: (**a**) Zr_40_Nb_35_Ti_25_, (**b**) Zr_50_Nb_35_Ti_15_, and (**c**) Zr_60_Nb_35_Ti_5_.

**Figure 7 materials-15-06565-f007:**
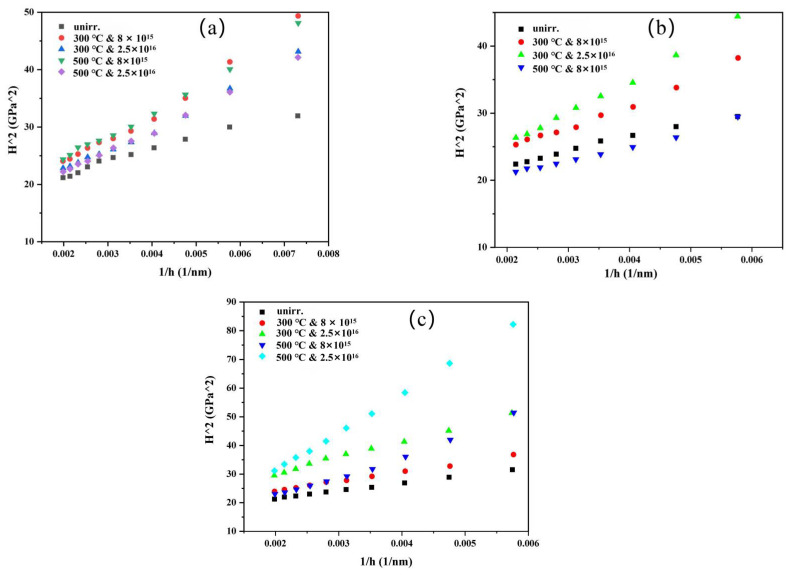
Nix–Gao model for three MEAs before and after Fe^11+^ irradiation: (**a**) Zr_40_Nb_35_Ti_25_, (**b**) Zr_50_Nb_35_Ti_15_, and (**c**) Zr_60_Nb_35_Ti_5_.

**Figure 8 materials-15-06565-f008:**
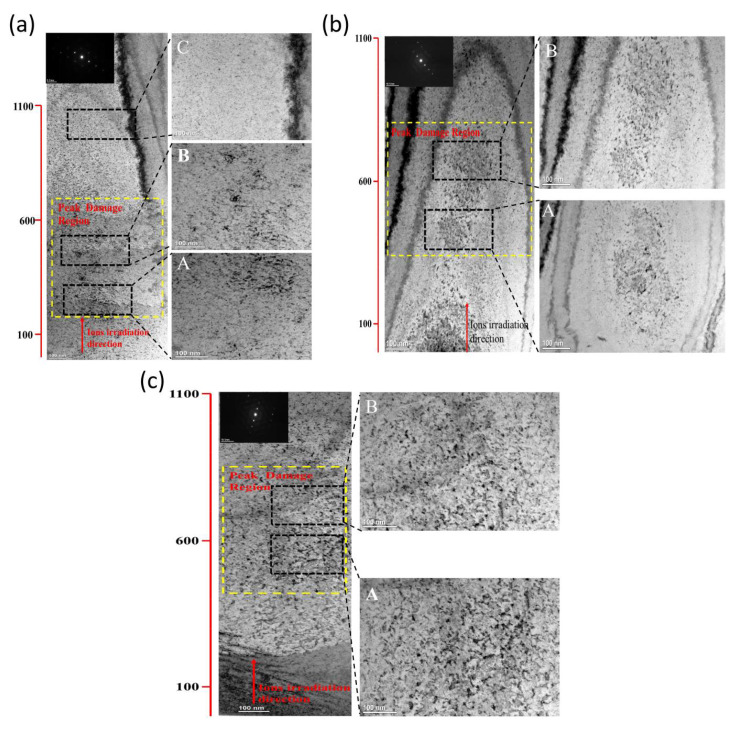
BF-TEM images of irradiated samples showing the irradiation-induced black-dot defects (**a**) 500 °C—8 × 10^15^ sample of Zr_40_Nb_35_Ti_25_, (**b**) 500 °C—8 × 10^15^ sample of Zr_50_Nb_35_Ti_15_, and (**c**) 500 °C—2.5 × 10^16^ sample of Zr_60_Nb_35_Ti_5_. The g = <110> near the [111] zone axis was selected. A, B, and C are the local enlargements of each sample along the irradiation direction.

**Figure 9 materials-15-06565-f009:**
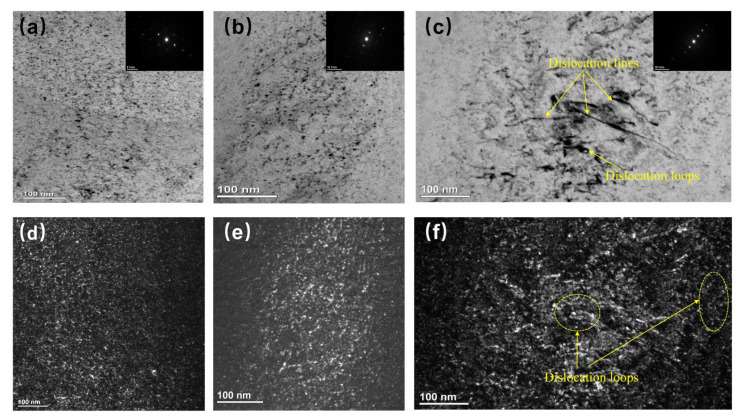
Enlarged BF and DF TEM images of Zr_40_Nb_35_Ti_25_ samples irradiated by 3 MeV Fe^11+^ ions: (**a**) 500 °C—8 × 10^15^, (**b**) 300 °C—2.5 × 10^16^, and (**c**) 500 °C—2.5 × 10^16^. (**d**–**f**) The corresponding weak-beam DF images. The g = <110> near the [111] zone axis was selected.

**Table 1 materials-15-06565-t001:** Nominal composition (at. %) of the three HEAs.

Sample No.	HEAs	Zr	Nb	Ti
1#	Zr_40_Nb_35_Ti_25_	45.06	40.16	14.78
2#	Zr_50_Nb_35_Ti_15_	53.46	38.12	8.42
3#	Zr_60_Nb_35_Ti_5_	61.05	36.28	2.67

**Table 2 materials-15-06565-t002:** The irradiation conditions of 3 MeV Fe^11+^ for the three HEAs, including temperature, fluence, and peak displacement damage, and peak dose rate calculated by SRIM 2018.

Samples	Temperature (°C)	Irradiation Fluence (ions/cm^2^)	Peak Damage (dpa)	Peak Dose Rate (dpa/s)
1#/2#/3#	300	8 × 10^15^	15	~6.5 × 10^−4^
		2.5 × 10^16^	47	
	500	8 × 10^15^	15	
		2.5 × 10^16^	47	

**Table 3 materials-15-06565-t003:** Calculation of relevant parameters of the Zr_40_Nb_35_Ti_25,_ Zr_50_Nb_35_Ti_15_, and Zr_60_Nb_35_Ti_5_ high-entropy alloys.

Alloys	ΔH_mix_ (KJ/mol)	ΔS_mix_ (J/mol·K)	Ω	δ (%)	VEC	Δχ	T_m_ (K)
Zr_40_Nb_35_Ti_25_	2.94	8.98	7.009	4.18	4.35	0.122	2294.74
Zr_50_Nb_35_Ti_25_	3.22	8.302	5.98	4.23	4.35	0.127	2317.8
Zr_60_Nb_35_Ti_5_	3.5	6.848	4.57	4.11	4.35	0.129	2336.5

**Table 4 materials-15-06565-t004:** The yield strength (σ_y_), ultimate tensile (σ_u_), and elongation after fracture (ε_ef_) for the ZrNbTi MEAs.

Alloys	Temperature (°C)	Yield Strength (δ_y_, MPa)	Ultimate Tensile (δ_u_, MPa)	Elongation after Fracture (ε_ef, %_)
Zr_40_Nb_35_Ti_25_ (1#)	Room	945	1010	17.5
Zr_50_Nb_35_Ti_15_ (2#)		903	915	14
Zr_60_Nb_35_Ti_5_ (3#)		1028	1050	11
Zr_40_Nb_35_Ti_25_ (1#)	400	350	423	28
Zr_50_Nb_35_Ti_15_ (2#)		355	408	25
Zr_60_Nb_35_Ti_5_ (3#)		360	390	20.5

**Table 5 materials-15-06565-t005:** Average nanohardness and hardening rate of Zr-Nb-Ti MEAs in the range of 200–400nm from the surface under different irradiation conditions.

Alloys	Zr_40_Nb_35_Ti_25_(GPa)	Zr_50_Nb_35_Ti_15_(GPa)	Zr_60_Nb_35_Ti_5_(GPa)	Hardening Rate (%)		
				Zr_40_Nb_35_Ti_25_	Zr_50_Nb_35_Ti_15_	Zr_60_Nb_35_Ti_5_
Unirradiated	5.17	5.15	5.22	——	——	——
8 × 10^15^/300 °C	5.56	5.53	5.62	7.54	7.38	7.66
2.5 × 10^16^/300 °C	5.48	5.84	6.42	6.01	13.40	22.99
8 × 10^15^/500 °C	5.66	4.95	5.9	9.48	−3.88	13.03
2.5× 10^16^/500 °C	5.43	——	7.5	5.03	——	42.72

## Data Availability

Not applicable.

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
