# Peer review of "A Novel BCC-Structure Zr-Nb-Ti Medium-Entropy Alloys (MEAs) with Excellent Structure and Irradiation Resistance"

_materials, 2022, doi:10.3390/ma15196565_

Round 1

Reviewer 1 Report

The authors need to sort out the gaps. For example, there are no spaces before the brackets (line 62, 62 and so on).

line 88

Maybe "TEM portrayal" or "TEM image"

Figure 2 shows optical images, in my opinion. Not from SEM.

The figures and captions are very small. It is difficult to make out anything.

The tables have moved to other pages.

You need to carefully review the names of the tables, for example, Table 4.

Paragraph 3.2. remained in paragraph 3.1.

It is required to finalize the article.

Do the authors write about HEAs or about MEAs?

Reviewer 2 Report

1.      The abstract is mainly qualitative. Some significant quantitative results should be added.

2.      Please check the grammatical errors such as “Fig.1 shown’ or “indicates the lattice constant gradually with the increase of Zr “ and so on.

3.      The quality of graphs (such as Figs.1, 2a, 5, 6, 7) is not good. The font size, legends, and numbers are too small.

4.      The authors should enlarge SEM and TEM images to be clearer.

5.      The author should mention the crosshead or initial strain rate during the tensile test.

6.      How did the authors calculate or measure the melting temperature of the alloys?

7.      It is suggested that the stress-strain curves at room and high temperature are plotted as either engineering or true.

8.      How did the author measure the hardness rate?

9.      The conclusions should be more comprehensive.

Reviewer 3 Report

The paper studies the mechanical properties, irradiation resistance, and irradiation hardening of three Zr-Nb-Ti alloys. The experimental design is sound, and the execution is comprehensive. However, there are still quite some issues that need revision. See my comments below:

1. Intro: “has gotten a tone of consideration lately” should be “ton”.

2. Intro: “We designed a novel Zr-Nb-Ti MEAs with BCC structure and low thermal neutron absorption cross-section and high thermal conductivity in this paper.” I feel like more introduction is needed here, especially regarding the Zr-base alloy. How is MEA expected to differ compared to traditional Zr-alloy? How low is the neutron absorption cross-section? And what is the thermal conductivity of this MEA? Any design consideration going into varying Zr and Ti content while keeping Nb content unchanged among 3 alloys? A lot of the background information is not mentioned here.

3. Exp: for the irradiation, what type of beam was used? Raster-scanned or defocused beam?

4. Exp: Figure 1, what’s the fluence of Fe ions that results in these SRIM plots? Also, why was the dpa axis scale very different comparing #1,2 and #3? Please double check. They should be compared with equal fluence. Table 2: what depth is associated with the reported dpa level?

5. Exp: “Thusly, the hardness at an indenter profundity” should be “Thus”.

6. Exp: “Since it is regularly realized that the space hardness is impacted by districts that are a few times the profundity of space, the space progress profundity between the close surface particle harmed layer and hidden flawless substrate not entirely settled to be 400nm for all light circumstances utilizing the strategy illustrated by Nix and Gao[28].” I’m very confused by what the authors are trying to say here. Please use terminologies in the field to explain and clarify. For example, I don’t know what “profundity” or “district” means?

7. Results: Figure 5&6, please enlarge the font of the legend. It’s quite hard to read the legend in the figures.

8. Results: Authors should obtain the statistics of the dislocation loops in terms of the type and density, and their relationship with hardening after irradiation. Recent studies by EDDE led by ORNL on Ni-based concentrated solid solution alloys demonstrate that the loop type, size, and density all have strong effects on radiation hardening. Can authors discuss the radiation hardening results from this perspective? 

9. Results: In terms of the TEM characterization of dislocation loops, what is the g vector used? Near which zone axis? Have authors done any Burgers vector analysis? If not, the TEM image can be extremely deceiving, because it may not truly present the dislocation population in the microstructure due to the invisibility criterion. One option is to use the two-beam condition with abovementioned conditions included, and another option is to use the recently developed on-zone S/TEM method to fully characterize all dislocation loop populations. Either way, I want to make sure these TEM images must be generated using comparable imaging conditions to draw meaningful conclusions when comparing three alloys.

Round 2

Reviewer 2 Report

The authors have addressed the issues. It is suitable for publication. However, I prefer the old version of the conclusions because they are more complete than the new version.

Author Response

Thank you for your commonts, the conclusions have been changed to the old version.

Reviewer 3 Report

The revision reached satisfaction for publication on Materials.

Author Response

Thank you for your comments. The English language has been improved.